# Presume Why Probiotics May Not Provide Protection in Inflammatory Bowel Disease through an Azoxymethane and Dextran Sodium Sulfate Murine Model

**DOI:** 10.3390/ijms23179689

**Published:** 2022-08-26

**Authors:** Ming-Luen Hu, Wei-Shiung Lian, Feng-Sheng Wang, Chao-Hui Yang, Wan-Ting Huang, Jing-Wen Yang, I-Ya Chen, Ming-Yu Yang

**Affiliations:** 1Graduate Institute of Clinical Medical Sciences, College of Medicine, Chang Gung University, Taoyuan 333, Taiwan; 2Division of Hepato-Gastroenterology, Department of Internal Medicine, Kaohsiung Chang Gung Memorial Hospital and Chang Gung University College of Medicine, Kaohsiung 833, Taiwan; 3Department of Medical Research, Kaohsiung Chang Gung Memorial Hospital and Chang Gung University College of Medicine, Kaohsiung 833, Taiwan; 4Department of Otolaryngology, Kaohsiung Chang Gung Memorial Hospital and Chang Gung University College of Medicine, Kaohsiung 833, Taiwan; 5Department of Laboratory Medicine, Kaohsiung Chang Gung Memorial Hospital and Chang Gung University College of Medicine, Kaohsiung 833, Taiwan; 6Department of Pathology, Kaohsiung Chang Gung Memorial Hospital and Chang Gung University College of Medicine, Kaohsiung 833, Taiwan

**Keywords:** inflammatory bowel disease, azoxymethane, dextran sodium sulfate, gut microbiota, probiotics, dysbiosis

## Abstract

Recent studies have shown dysbiosis is associated with inflammatory bowel disease (IBD). However, trying to restore microbial diversity via fecal microbiota transplantation (FMT) or probiotic intervention fails to achieve clinical benefit in IBD patients. We performed a probiotic intervention on a simulated IBD murine model to clarify their relationship. IBD was simulated by the protocol of azoxymethane and dextran sodium sulfate (AOM/DSS) to set up a colitis and colitis-associated neoplasm model on BALB/c mice. A single probiotic intervention using *Clostridium butyricum Miyairi* (CBM) on AOM/DSS mice to clarify the role of probiotic in colitis, colitis-associated neoplasm, gut microbiota, and immune cytokines was performed. We found dysbiosis occurred in AOM/DSS mice. The CBM intervention on AOM/DSS mice failed to improve colitis and colitis-associated neoplasms but changed microbial composition and unexpectedly increased expression of proinflammatory IL-17A in rectal tissue. We hypothesized that the probiotic intervention caused dysbiosis. To clarify the result, we performed inverse FMT using feces from AOM/DSS mice to normal recipients to validate the pathogenic effect of dysbiosis from AOM/DSS mice and found mice on inverse FMT did develop colitis and colon neoplasms. We presumed the probiotic intervention to some extent caused dysbiosis as inverse FMT. The role of probiotics in IBD requires further elucidation.

## 1. Introduction

With the advances of next-generation sequencing, research focusing on gut microbiota has thrived over the past years. The number of microorganisms inhabiting the gastrointestinal (GI) tract has been estimated to exceed 10^14^, and the genomic contents of microorganisms are 100 times more than the human genome [1,2]. Maintaining the balance of the gut microbiota is crucial for keeping the gut in good health because short-chain fatty acids (SCFAs) from the metabolites of intestinal bacteria play many important physiologic roles in protecting the gut environment [3]. SCFAs include acetic acids, propionic acids, and butyric acids. Especially butyric acids help intestinal barrier function and immune system through G-protein-coupled receptors and regulate histone deacetylase activity [4]. They also act as energy substrates for colonocytes and intestinal bacteria. The imbalance of the intestinal microbiota will decrease the amount of SCFAs and induce colitis [5,6,7].

Inflammatory bowel disease (IBD) is a chronic inflammatory disease of the GI tract, which mainly contains Crohn’s disease and ulcerative colitis. Many studies have revealed the possible pathogenesis is associated with host genetic susceptibility, environmental factors, and immunological abnormalities. Recent studies have shown that decreased microbial diversity or dysbiosis is also associated with IBD [8,9,10,11]. Therefore, restoration of the microbial diversity and improved gut microbiota may be a treatment strategy in IBD patients.

Long-acting inflammation and cancer development are known to be intertwined. Patients with long-term IBD have a higher risk of developing colorectal cancer (CRC) [12]. Several animal models in search of the detailed mechanism of colitis-associated CRC (CAC) have been developed. Of them, a mouse model by the protocol of using azoxymethane (AOM) in combination with dextran sodium sulfate (DSS) is commonly used [13].

Fecal microbiota transplantation (FMT) is a rising topic and refers to the administration of intestinal microbes from a healthy donor into a recipient with the intent of modifying the recipient’s intestinal microbiome via oral delivery or colonoscopy [14]. The aim of FMT is to restore the balance and diversity of gut microbiota to keep the gut of recipients in good health. In patients with recurrent or refractory *Clostridium difficile*-associated colitis, FMT brings amazing treatment effects [15,16,17]. For FMT in IBD patients, FMT increases the diversity of gut microbiota and should be effective in IBD patients.

Probiotics are those live microorganisms which produce SCFAs in the gut. The association of probiotics with well-being has a long history, as it has been more than one century since it was observed that gut microbiota from healthy breast-fed infants were dominated by *Bifidobacteria* which were absent from formula-fed infants suffering from diarrhea, establishing the concept that probiotics played a role in maintaining good health [18]. With the promotion of food industries, many probiotics are available in the market and widely used due to the benefits in host gut protection. However, related research about probiotics to repair mucosal damage and improve gut inflammation in IBD patients is not promising. The routine use of probiotics in IBD patients is not recommended in the current treatment guidelines [19,20].

*Clostridium butyricum Miyairi* (CBM) is an anaerobic spore-forming Gram-positive bacterium. It was first isolated in Japan in 1933 and has been commonly used as a probiotic strain for a long time [21]. CBM is resistant to the acid environment of the stomach and can smoothly enter the small intestine and colon. It produces large amounts of SCFAs to protect the gut environment and has been shown to have many benefits in the prophylaxis of bacteria or antibiotics-associated diarrhea [22,23].

In theory, probiotics and FMT are feasible options to treat or improve the gut health of IBD and decrease colitis and subsequent CAC. However, the benefit of probiotics in the clinical response of IBD patients are not proven. In addition, the benefit of FMT in IBD patients is not attractive and is only limited in the induction of ulcerative colitis remission [24]. Thus, the roles and benefits of probiotics and FMT in the host gut environment of IBD patients warrant further elucidation.

Our study aim: we hypothesized probiotics or FMT in the condition of severe colitis with loss of mucosal barrier function such as moderate or severe IBD could not adhere to mucosa to provide protection in gut health. To simplify the reciprocal interactions between transplanted microbiomes and host gut and to avoid the complex and uncertain microbiomes of FMT, we tried to intervene only on one probiotic strain using CBM on AOM/DSS mice to observe and analyze the probiotic role in gut pathology, gut microbiota, and host immunity on a simulated IBD model.

## 2. Results

### 2.1. Establish a Simulated IBD Model of Colitis and Colitis-Associated Neoplasm by Using AOM/DSS Protocol on Mice

We were able to replicate a colitis and colitis-associated neoplasm murine model in specific pathogen-free (SPF) BALB/c mice by using an AOM/DSS protocol (Figure 1A). The body weight gain in AOM/DSS group was significantly slower than that in the control group (Figure 1B). The FOBT began to turn positive in the AOM/DSS group in the third week of the protocol but not in the control group (Figure 1C). Colitis and colon neoplasms developed in the AOM/DSS group, but not in the control group (Figure 1D). Microscopic examinations revealed inflammation or neoplasms in the rectum and colon in the AOM/DSS group but not in the control group (Figure 1E).

### 2.2. Confirm Dysbiosis Occurrence on AOM/DSS Mice with Colitis

To confirm if dysbiosis occurred on AOM/DSS mice, we analyzed gut microbiota between AOM/DSS and control groups. Weighted PCoA (Figure 2A) and unweighted PCoA (Figure 2B) analysis revealed different β diversity between the two groups including sequence distances and abundance (*p* = 0.0475 and *p* = 0.0335, respectively). Chao1 boxplot (Figure 2C) and Shannon boxplot (Figure 2D) showed different α diversity between the two groups (*p* = 0.0151 and *p* = 0.0052, respectively). The expressions of the four major phyla between the two groups are shown in Figure 2E, and a different abundance of these four phyla was observed. Different abundances of each family in Bacteroidetes and Firmicutes between the two groups were also observed (Figure 2F). In the analysis of fecal microbial composition, Venn diagram (Figure 2G) and ANCOM (Figure 2H) results revealed different microbial composition and abundance between the two groups. Cladogram (Figure 2I) and LDA score (Figure 2J) selected different and predominant bacteria between the two groups. The fecal microbiomes between the two groups go on different metabolic pathways especially in the secretion system, and glyoxylate and dicarboxylate metabolism (Figure 2K). All the results indicated decreased microbial diversity and dysbiosis occurred on AOM/DSS mice.

### 2.3. Probiotic Intervention Failed to Inhibit Colitis or Colitis-Associated Neoplasms on AOM/DSS Mice

We added CBM intervention (5.0 × 10^7^ CFU) to the AOM/DSS mice via oral gavage twice a week after each round of DSS administration (four times/round) for three rounds (Figure 3A) to investigate if probiotics could inhibit the development of colitis and colitis-associated neoplasms. We found the body weight gain of mice did not differ between the AOM/DSS + CBM group and AOM/DSS group (Figure 3B). At the third week of the protocol, FOBT began to show positivity in both groups. The scores of FOBT were higher in the AOM/DSS + CBM group than in the AOM/DSS group at the 4th week and 6th week of the protocol (Figure 3C). Gross appearance of the colon showed colitis and neoplasms in both groups and the number of neoplasms in the AOM/DSS + CBM group was more than those in the AOM/DSS group (*p* = 0.047, Figure 3D). Microscopic examination showed inflammation and neoplasms in the rectum and colon in both groups (Figure 3E). These observations demonstrated CBM intervention did not improve colitis or prevent colitis-associated neoplasms in AOM/DSS mice.

### 2.4. Probiotic Altered Gut Microbiota but Failed to Improve Dysbiosis

To further explore the impact of CBM intervention on dysbiosis, we analyzed gut microbiota between the AOM/DSS + CBM group and AOM/DSS group. PCoA analysis revealed different β diversity between the two groups (Figure 4A). The heatmap also showed different abundances of microbiomes between the two groups (Figure 4B). The ratio of Firmicutes/Bacteroidetes between the two groups was slightly different (Figure 4C). Venn diagram (Figure 4D) and ANCOM (Figure 4E) results revealed different microbial composition between the two groups. Aerobic expression of fecal microbiota was lower in the AOM/DSS + CBM group (*p* = 0.0318, Figure 4F) but no difference in anaerobic expression (Figure 4G). Cladogram (Figure 4H) and LDA score (Figure 4I) selected different and predominant bacteria between the two groups. In addition, the fecal microbial composition in the AOM/DSS + CBM group was more potentially pathogenic than that in the AOM/DSS group (*p* = 0.0036, Figure 4J). Figure 4K shows the expression of each metabolic pathway in the microbiomes from the two groups. All these results revealed the CBM intervention altered gut microbiota but failed to improve dysbiosis. Instead, it caused the effect of dysbiosis to some extent.

### 2.5. Dysbiosis Is Also the Contributing Cause of Colitis through an Inverse FMT Model

The murine AOM/DSS model successfully evoked colitis and colitis-associated neoplasms and showed dysbiosis. It was certain that AOM and DSS are causative agents of colitis and colon neoplasms. However, it was not clear whether the dysbiosis caused by AOM and DSS is also a contributing factor to colitis. If dysbiosis induces colitis and colitis-associated neoplasm, that will explain why CBM intervention fails to improve colitis due to dysbiosis effects. To clarify this issue, we performed inverse FMT. Inverse FMT was defined as FMT using feces from AOM/DSS mice with dysbiosis to healthy recipient mice. In the FMT group, the four recipient mice received fecal enema from AOM/DSS mice during the third round of DSS, three times per week for three weeks (Figure 5A). These FMT mice grew significantly slower than mice in the control group (*p* < 0.01; Figure 5B) and showed positive FOBT at the 7th week (Figure 5C). Gross appearance of the colon showed colitis and neoplasms polyps in the mice receiving FMT (Figure 5D). Microscopic examinations revealed inflammation and neoplasms in the rectum and colon of the FMT recipients (Figure 5E). All the findings disclosed that dysbiosis via inverse FMT causes colitis and neoplasm development, thereby supporting CBM intervention with the effect of dysbiosis.

### 2.6. Probiotic Intervention Increased Pro-Inflammatory Cytokine Expression

The inflammatory cytokine is an important index for evaluating the severity of colitis, so we screened the expressions of pro-inflammatory cytokines (*IL-1β*, *IL-6*, *IL-17*, and *TNFα*) and anti-inflammatory cytokines (*IL-10* and *TGFβ*) using qPCR analysis in colon tissues of mice between control and AOM/DSS groups. We found the expression of *IL-17* was significantly up-regulated in AOM/DSS group and the expressions of the other five cytokines were not different (Figure 6A). To further validate the protein expression of IL-17A, we performed the IL-17A IHC staining among control, AOM/DSS, inverse FMT, and AOM/DSS + CBM groups and calculated the IL-17A-positive areas. As shown in Figure 6B, the expressions of IL-17A in both rectum and colon were significantly higher in the three experimental groups than the control group (all *p* < 0.05). In the AOM/DSS + CBM group, the expression of IL-17A was significantly higher than that in AOM/DSS group in the rectum (*p* < 0.001, Figure 6B). These results revealed the CBM intervention in AOM/DSS- induced colitis increased inflammation with the effect of dysbiosis.

## 3. Discussion

IBD is a chronic inflammatory disease of the GI tract. Not only does it deteriorate life quality and social activities of patients, but also harbors the risk of CRC development. Recent studies revealed the incidence had been increasing over the past decades [25]. The pathogenesis of IBD is associated with genetics, environment, immune reactions, and gut microbiota. Regarding the gut microbiota in IBD, many studies revealed dysbiosis and decreased microbial diversity in IBD patients [11,26]. The phyla of *Firmicutes* and *Bacteroidetes* are decreased in IBD patients, whereas the phyla of *Actinobacteria* and *Proteobacteria* are relatively increased. More specifically, adherent invasive *Escherichia coli, Pasteurellaceae, Veillonellaceae, Fusobacterium* species, and *Ruminococcus gnavus* are increased. Otherwise, *Clostridium* groups IV and XIVa, *Bacteroides*, *Suterella*, *Roseburia*, *Bifidobacterium* species and *Faecalibacterium prausnitzii* are decreased in IBD patients [27,28]. In our study, we successfully established the AOM/DSS murine model that simulated human IBD. Mice on the AOM/DSS protocol developed colitis and colitis-associated neoplasms from the normal colon. We found *Verrucomicrobiales* and *Akkermansiaceae* were predominant in the control group. Otherwise, *Bacilli* and *Actinobacteria* were predominant on AOM/DSS mice. Similar with IBD patients, altered gut microbiota were found on AOM/DSS mice developing colitis.

From the pathophysiological point of view, altered gut microbiota in IBD can be explained due to the increase of some bacteria which induce some proinflammatory cytokines and the reduction of other bacteria which induce anti-inflammatory cytokines and product beneficial metabolites, thereby disturbing the gut immune homeostasis, and decreasing the gut protection. Thus, the treatment effect should be achieved if the intervention of good live microorganisms can be adhering to the intestinal epithelium, producing beneficial metabolites, stabilizing the intestinal microbiota, and stimulating anti-inflammatory cytokines in the gut environment [29].

Probiotics are good live microorganisms which produce SCFAs to improve the gut environment. Therefore, we treated AOM/DSS mice with probiotic CBM and hoped to improve the condition of colitis. In our study design, only one strain and non-preconditioning probiotic intervention was administered intermittently on AOM/DSS mice. No other anti-inflammatory medical treatment was given in additional to CBM. We found the abundances of *Pseudomonadaceae*, *Lactobacillaceae*, *Clostridiaceae_1*, were decreased and the abundances of *Aerococcaceae*, *Ruminococcaceae*, *Saccharimonadaceae* were increased on CBM-treated AOM/DSS mice. We found CBM intervention altered the gut microbiomes but did not improve dysbiosis or decrease the severity of colitis and colitis-associated neoplasms on these CBM-treated AOM/DSS mice. Instead, the proinflammatory cytokine IL-17A was increased on these CBM-treated AOM/DSS mice.

Inconsistent with the results of our study, two studies showed *Clostridium butyricum* reduced colitis-associated CRC on AOM/DSS murine models and decreased proinflammatory cytokines TNF-α, IL-6, and COX-2, and increased anti-inflammatory cytokine IL-10 [30,31]. In addition to CBM on the AOM/DSS model, a study from Silveira et al. showed another strain intervention using *Lactobacillus bulgaricus* on AOM/DSS mice inhibited tumor volume and decreased pro-inflammatory cytokines IL-6, TNF-α, IL-17, IL-23 and IL-1β [32]. Another study from Wang et al. showed multi-strain probiotic intervention using VSL#3, which contained eight probiotic strains including *Lactobacillus paracasei*, *Lactobacillus plantarum*, *Lactobacillus acidophilus*, *Lactobacillus delbrueckii subsp.*
*bulgaricus*, *Bifidobacterium longum*, *Bifidobacterium*
*breveCM*, *Bifidobacterium infantis*, and *Streptococcus*
*thermophilus*, altered the gut microbiomes and reduced the tumor load of AOM/DSS mice. In their study, they also used an anti-inflammatory medicine, 5-ASA, for further analysis. They found both AOM/DSS mice treated with VSL#3 plus 5-ASA and VSL#3 alone improved colitis, but without head-to-head comparison between these two groups [33]. In addition, some studies showed different probiotic doses causing different effects. A study from Sha et al. showed pre-administered and low dose of probiotic intervention (10^7^ CFU/day) using *Escherichia coli* Nissle 1917 on trinitrobenzene sulfonic acid (TNBS)-treated mice significantly improved colitis. However, a pre-administered and high dose (10^9^ CFU/day) failed to improve colitis and actually caused deterioration [34]. Another study from Komaki et al. showed *Lactococcus lactis* intervention on DSS mice deteriorated colitis, increased IFN-γ, TNF-α and IL-6. Furthermore, mice on higher doses tended to have decreased survival [35]. The results of the probiotic interventions on mouse colitis models are summarized in Table 1. Thus, the conflicting and uncertain results about the benefit of probiotic intervention require clarification.

Nowadays, accumulating evidence has shown *Akkermansia muciniphila* (*A. muciniphila*) plays an important role in gut health. *A. muciniphila*, colonizing the human gut and accounting for 3–5% of the gut microbiome, was first isolated in 2004 by Derrien et al. and has been found with various benefits in obesity, diabetes mellitus and metabolic syndrome in addition to gut health [36,37]. Some studies revealed lower colonization and abundance of *A. muciniphila* in IBD patients and mouse colitis models. Intriguingly, either live or pasteurized *A. muciniphila* improved colitis in mice [38,39]. *A muciniphila* has been considered a promising probiotic strain in the future and is worth further study.

From our inverse FMT study, the normal recipient mice receiving the feces from mice on 3rd round DSS subsequently developed colitis and neoplasms, which means fecal transplantation with dysbiotic gut microbiomes may lead to the development of colitis and colitis-associated neoplasms in recipient guts. Perhaps inverse FMT explained why CBM intervention changed gut microbiota and increased proinflammatory cytokines from the effect of dysbiosis. The aim of FMT is to restore the balance and diversity of gut microbiota to keep the gut in good health. However, FMT with harmful microbial components may lead to colitis and neoplasms. Recently, cases with bacteremia and mortality from FMT were reported [40]. Therefore, how to screen and define the standard and healthy stool for fecal transplantation is important. However, the potential risks of FMT such as occult infection, inflammation or cancer development have not been excluded to date. Moreover, one should be concerned about the reciprocal interaction between host immunity and transplanted microbiomes.

Probiotics are generally considered helpful for the gut in good health and not harmful even with long term and daily use. However, from our study, we are not sure if probiotics always keep the gut in good health. In specific conditions, they may fail to improve colitis or even increase gut inflammation. The detailed mechanisms are necessary to be explored. It is possible that different durations, frequencies, doses of intervention, or different strains of probiotics will bring different results. Severe colitis with fragile mucosa and decreased integrity of mucosal barrier may not keep probiotics adhering to mucosa to produce enough SCFAs. Moreover, probiotic intervention may be even regarded as the invading microorganism by host immunity and induce a series of immune responses. Furthermore, the competitive or reciprocal interactions with other existing gut microorganisms may aggravate gut inflammation. Perhaps they can explain these conflicting results about the benefit of probiotic intervention on mice with severe colitis.

Gut microbiota plays a vital role in our gut health, but it remains to be explored well regarding reciprocal interactions between gut microbiomes and host immunity. Using simple and specific probiotics instead of complex FMT seems to be a practical and regulable option. Unfortunately, the treatment benefits of probiotics in IBD patients are limited. Using probiotics as a treatment option is still not included in the current treatment guidelines [41,42,43,44,45]. FMT or probiotics are helpful in our gut health but they may cause dysbiosis to some extent and do some harm in our gut, especially in hosts with severe colitis. Therefore, we suggest probiotic interventions in patients with severe colitis or in IBD patients with active gut inflammation should be used more carefully. More studies are necessary to clarify their roles among gut pathology, gut microbiota, and host immunity.

## 4. Materials and Methods

### 4.1. Study Design

Due to fact that the benefit of probiotics in IBD was not demonstrated, we hypothesized probiotics in the condition of severe colitis with loss of mucosal barrier function such as moderate or severe IBD could not improve dysbiosis and provide protection in gut health. First, we set up a murine model of colitis and colitis-associated neoplasms through the protocol of AOM/DSS to simulate IBD and analyzed gut microbiota on mice with AOM/DSS-induced colitis. Second, we performed a probiotic intervention using CBM on AOM/DSS mice to observe if there was an improvement of colitis and colon neoplasms. We analyzed if CBM intervention altered the change of microbial composition and decreased correlative inflammatory cytokines to clarify the role of CBM intervention in IBD. If CBM intervention improved colitis and colon neoplasms, we would try to adjust the dose of CBM to achieve better treatment effect. If the CBM intervention failed to improve colitis or colon neoplasms, we would try to analyze the possible mechanism.

### 4.2. AOM/DSS Colitis and CRC Model

Four-week-old male BALB/c mice, weighing 18–20 g, were purchased from the National Laboratory Animal Center of Taiwan. All the animal use protocols were approved by the Institutional Animal Care and Use Committee of Kaohsiung Chang Gung Memorial Hospital (Approval No. 2020022501). Mice were maintained under SPF conditions and housed in groups of four or five animals per cage and kept under controlled 12 h light–dark cycle with free access to standard diet and water.

The protocol for induction of colitis/CRC (Figure 1A) was modified from previously reported studies [46,47,48]. After two weeks of adaptation, mice received a single intraperitoneal (*i.p.*) injection of AOM (Merck, Kenilworth, NJ, USA; 10 mg/kg of body weight) on day 1 followed by 3 rounds of DSS beginning on day 6. For each round of DSS, water containing 2% DSS (MP Biomedicals, Santa Ana, CA, USA) was administered to mice for 5 days followed by 16 days of water for recovery. For control group, mice received a single *i.p.* injection of phosphate-buffered saline (PBS) on day 1 and did not receive any DSS. During the protocol, we recorded body weight and performed fecal occult blood test (FOBT; Shih-Yung Medical Instruments, Taipei, Taiwan) on each mouse every day. The score of FOBT from zero to 4+ indicated negative, mild, moderate, and severe bleeding reaction in the feces. All the mice were euthanized after 3 rounds of DSS, i.e., day 68 of AOM/DSS protocol (Figure 1A). The CBM feces of all the mice were collected at the end of the protocol and stored at −80 °C immediately for analysis of gut microbiota. The colon tissues were collected for histological and real-time quantitative polymerase chain reaction (qPCR).

### 4.3. Microbiota Analysis

Fresh stool samples were collected and stored at −80 °C immediately and transferred to the Laboratory of Biotools (Taipei, Taiwan) for microbial analysis. Total genomic DNA from stool samples was extracted using the column-based Stool DNA Kit (CatchGene, New Taipei, Taiwan). For the 16S rRNA gene sequencing, V3-V4 hypervariable region was amplified according to the 16S Metagenomic Sequencing Library Preparation procedure (Illumina, San Diego, CA, USA) and sequenced on an Illumina MiSeq platform (Illumina).

### 4.4. Probiotic Intervention

Mice on AOM/DSS protocol were given a sporular preparation of CBM (5.0 × 10^7^ colony forming units (CFU)/0.3 mL sterilized water; Miyarisan^®^, Nagano, Japan) via oral gavage using an 8-cm stainless steel cannula, twice a week after DSS administration for 3 rounds, 12 times totally. For sham controls, mice were given sterilized water via oral gavage at the same time instead of CBM.

### 4.5. FMT

For ex vivo FMT, feces from donor mice were collected, pooled, and mixed daily and stored at −80 °C. Feces pellets (150 mg) collected in sterile tubes were homogenized in 1 mL of PBS and centrifuged at 2000× *g* at 4 °C for 1 min. The bacteria-enriched supernatants were collected and centrifuged for 5 min at 15,000× *g* and the bacterial pellets were resuspended in 1 mL of sterile saline as microbiota transplants. The recipient mice were treated with 0.3 mL of microbiota transplants 3 times per week for 3 weeks via enema.

### 4.6. qPCR Analysis for Expression of Cytokines

Colon tissue was homogenized, total RNA was isolated using the Nucleospin RNA kit (Macherey-Nagel, North Rhine-Westphalia, Germany) and cDNA was synthesized using an iScript kit (Bio-Rad, Hercules, CA, USA) according to the manufacturer’s protocols. The gene expression of *IL-1β*, *IL-6*, *IL-10*, *IL-17*, *TGFβ*, and *TNFα* was analyzed using a SYBR green system for qPCR. Sequences of forward and reverse primers and amplicon sizes of these genes and internal control *β-actin* (*ACTB*) gene are: *IL-1β* (5′-GAT GAT AAC CTG CTG GTG TGT GA-3′, 5′-GTG TTC ATC TCG GAG CCT GT AG-3′, 67 bp), *IL-6* (5′-CCA CCG GGA ACG AAA GAG AA-3′, 5′-GAG AAG GCA ACT GGA CCG AA-3′, 63 bp), *IL-10* (5′-AGC ATG GCC CAG AAA TCA AG-3′, 5′-CGC ATC CTG AGG GTC TTC AG-3′, 67 bp), *IL-17* (5′-TCT GTG TCT CTG ATG CTG TGC T-3′, 5′-ATC GCT GCT GCC TTC ACT GTA-3′, 62 bp), *TGFβ* (5′-CTG CTG CTT TCT CCC TCA AC-3′, 5′-CAC TAG AAG CCA CGG GAG TG-3′, 62 bp), *TNFα* (5′-CAC CGT CAG CCG ATT TGC-3′, 5′-TTG ACG GCA GAG AGG AGG TT-3′, 60 bp), and *ACTB* (5′-AATCGTGCGTGACATCAAAGAG-3′,5′- GCCATCTCCTGCTCGAAGTCTA-3′, 63 bp). The 10-μL reaction mix contained 50 ng cDNA, 200 nM each primer, and 5 μL 2× Power SYBR^®^ Green PCR Master Mix (Applied Biosystems, Waltham, MA, USA) were run in 7500 Fast Real-Time System (Applied Biosystems) using thermal cycling parameters 95 °C for 10 min, followed by 40 cycles of PCR reaction at 95 °C for 20 s and 60 °C for 1 min.

### 4.7. Histological Analysis and IL-17A Immunohistochemical (IHC) Staining

All the colons were harvested at necropsy, luminal contents were flushed, and cut open longitudinally to count and measure the dimension of tumors. The whole colon was divided into two sections, the rectum and colon, for microscopic examination. All the colons were then fixed and embedded in 4% paraffin. Three-micrometer sections were used for hematoxylin and eosin (H&E) staining. For IL-17A IHC staining, the tissue sections were incubated with polyclonal antibody against IL-17A (ab79056, Abcam, Cambridgeshire, UK) (1:200 dilution) overnight and a Novolink Polymer Detection Systems (RE7150-CE; Leica Biosystems, Richmond, IL, USA) was used to visualize the specific binding of the secondary antibody to the IL-17A antibody. All the H&E and IL-17A IHC staining slides were scanned using a whole slide scanner (Pannoramic MIDI; 3DHistech, Budapest, Hungary). The positive areas for IL-17A antibody staining were calculated using ImageJ free software (NIH, Bethesda, MD, USA).

### 4.8. Statistical Analysis

The differences of weight gain, FOBT, tumor number, microbial composition, and cytokine expression between two independent groups were analyzed by Mann-Whitney U test due to small sample size. Two-sided *p* value was calculated, and a difference was considered statistically significant if *p* value was <0.05. All computations were performed using SPSS for Windows Release 19.0 software (SPSS, Chicago, IL, USA) and Graph Pad Prism 7.04 (GraphPad, San Diego, CA, USA).

## 5. Conclusions

Probiotic intervention in IBD patients may not always provide protection. To some extent, it can cause dysbiosis and elicit further inflammation. There are still some limits in our study and it is worth investigating the treatment effects in different strains, durations, doses, frequencies of probiotic intervention, with or without precondition. The role of probiotics in IBD should be explored more and probiotics in IBD patients should be used more cautiously.

## Figures and Tables

**Figure 1 ijms-23-09689-f001:**
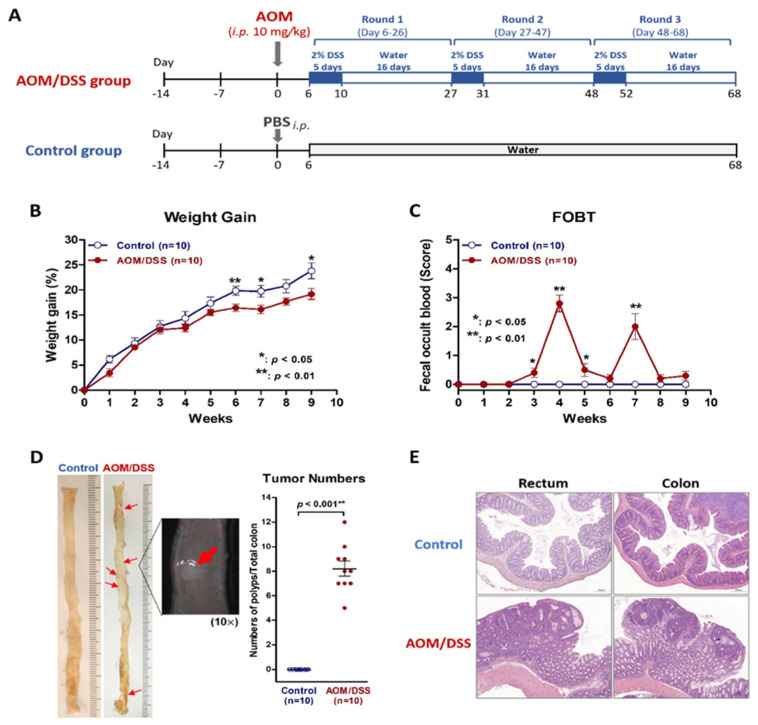
The murine colitis and colorectal cancer model using AOM/DSS on BALB/c mice. (**A**) The experimental AOM/DSS protocol for induction of colitis/colorectal cancer. (**B**) The weight gain of mice in the AOM/DSS group increased more slowly than that in control group. (**C**) Fecal occult blood turned positive since the third week of protocol in the AOM/DSS group but not in the control group. (**D**) The gross appearance of the whole colon in the AOM/DSS group showed colitis and developed colon neoplasms after the third round of DSS. (**E**) Microscopic examination in H&E staining showed inflammation and neoplasms in the colon in the AOM/DSS group but not in the control group. * *p* < 0.05; ** *p* < 0.01.

**Figure 2 ijms-23-09689-f002:**
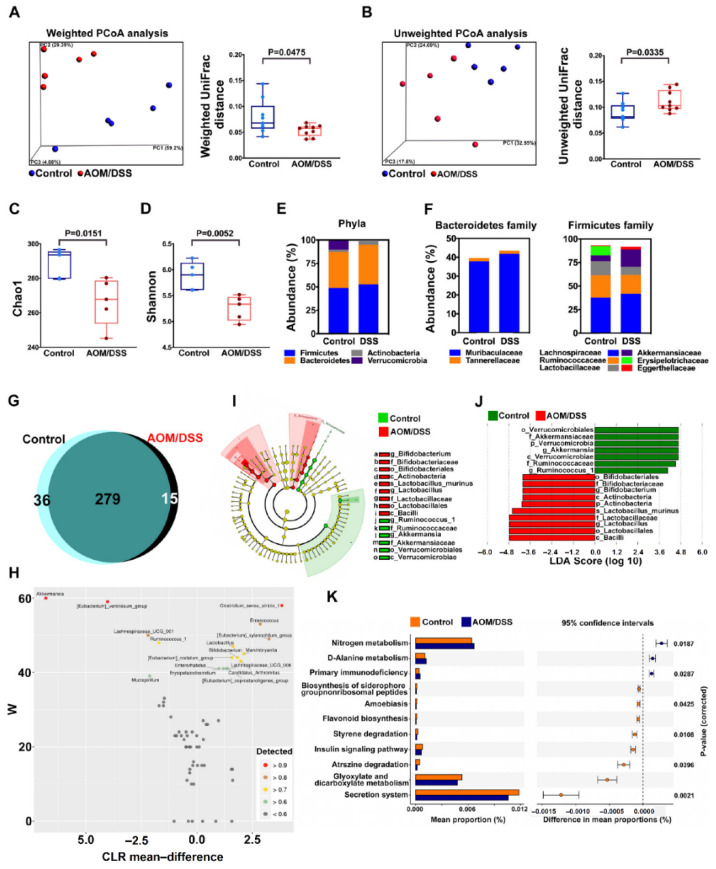
The analysis of fecal microbiota between the AOM/DSS group and control group. (**A**) Weighted and (**B**) unweighted PCoA analysis revealed different β diversity between the two groups including sequence distances and abundance. (**C**) Chao1 boxplot and (**D**) Shannon boxplot showed different α diversity between the two groups. (**E**) The expressions of the four major phyla between the two groups. (**F**) The expression of each family from Bacteroidetes and Firmicutes between the two groups. (**G**) Venn diagram and (**H**) ANCOM results revealed different microbial composition and abundance between the two groups. (**I**) Cladogram and (**J**) LDA score selected different and predominant bacteria between the two groups. (**K**) The fecal microbiomes between the two groups go on different metabolic pathways especially in the secretion system, glyoxylate and dicarboxylate metabolism, and nitrogen metabolism.

**Figure 3 ijms-23-09689-f003:**
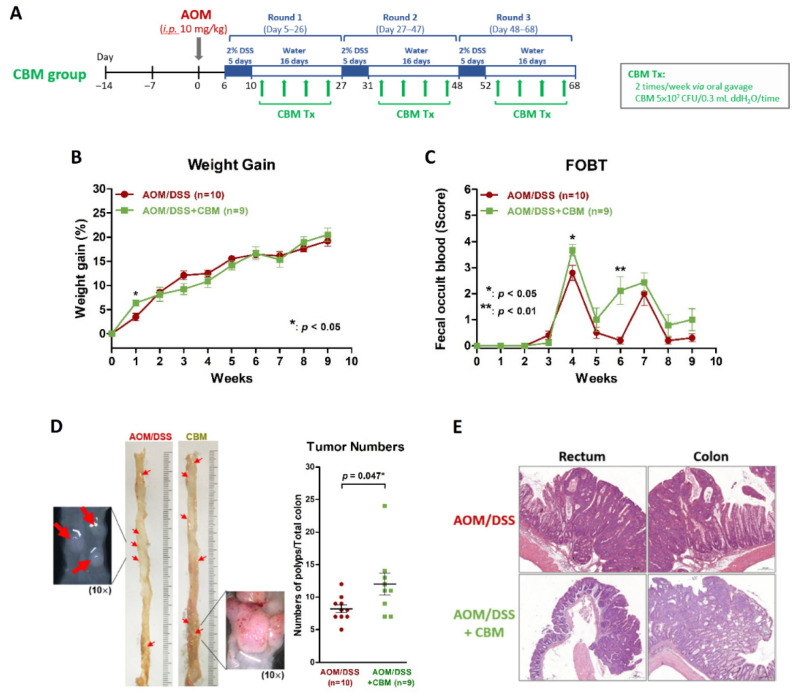
Probiotic intervention using CBM in a murine colitis and colitis-associated neoplasm model using AOM/DSS. (**A**) The protocol of the AOM/DSS + CBM model. (**B**) The body weight gain in the AOM/DSS + CBM group was not different from that in the AOM/DSS group during the second to the ninth week of protocol. (**C**) Fecal occult blood showed positivity in both groups since the third week of the protocol. At the fourth and sixth weeks, the score of fecal occult blood was significantly higher in the AOM/DSS + CBM group than that in the AOM/DSS group. (**D**) Gross appearance showed inflammation and neoplasms in the colon of both the AOM/DSS group and AOM/DSS + CBM group and the number of colorectal neoplasms in the AOM/DSS + CBM group was significantly higher than that in the AOM/DSS group. (**E**) Microscopic examinations in H&E staining showed inflammation and neoplasms in the colon in both the AOM/DSS and AOM/DSS + CBM groups. * *p* < 0.05; ** *p* < 0.01.

**Figure 4 ijms-23-09689-f004:**
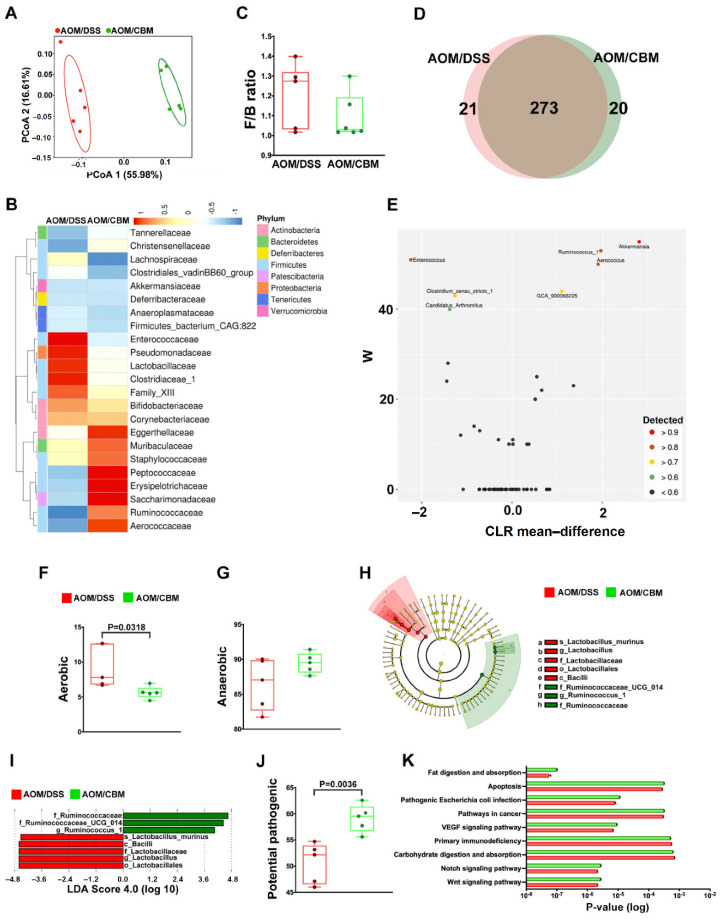
The analysis of fecal microbiota between the AOM/DSS + CBM group (AOM/CBM) and AOM/DSS group (AOM/DSS). (**A**) PCoA analysis revealed different β diversity between the two groups. (**B**) The heatmap showed different microbial compositions between the two groups. (**C**) F/B ratio showed the ratio of Firmicutes/Bacteroidetes between the two groups. (**D**) Venn diagram and (**E**) ANCOM results revealed different microbial compositions and abundances between the two groups. (**F**,**G**) Aerobic and anaerobic expressions of fecal microbiota between the two groups. (**H**) Cladogram and (**I**) LDA score selected different and predominant bacteria between the two groups. (**J**) The fecal microbial composition in AOM/DSS + CBM group showed more potentially pathogenic than that in AOM/DSS group. (**K**) The expression of each metabolic pathway in the microbiomes from the two groups.

**Figure 5 ijms-23-09689-f005:**
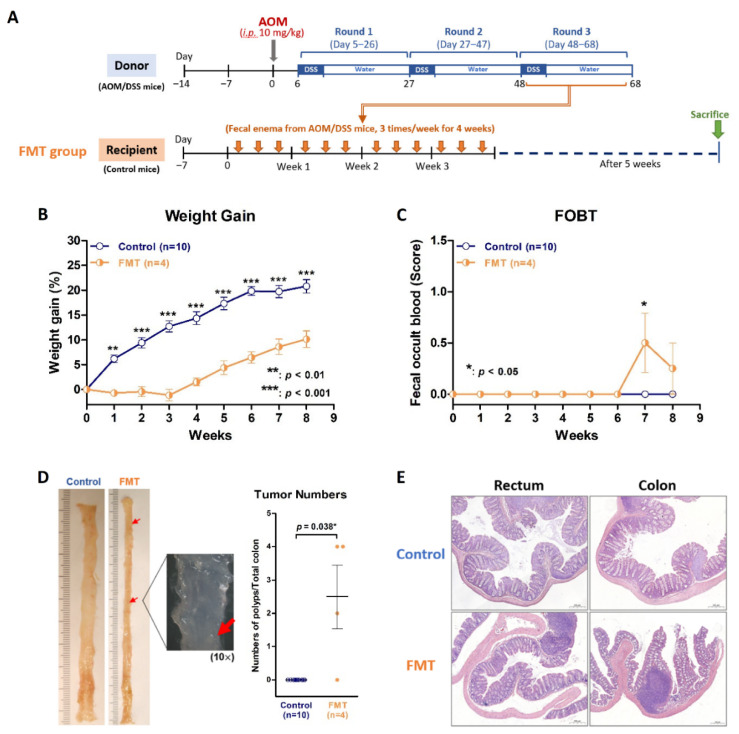
Inverse fecal microbiota transplantation (inverse FMT) from AOM/DSS mice to normal recipient mice. (**A**) The protocol of inverse FMT experiment. (**B**) The body weight gains were significantly lower in FMT recipient mice than those in control mice. (**C**) Fecal occult blood turned positive at the seventh week of protocol in FMT recipients. (**D**) Gross appearance showed inflammation and neoplasm polyps in the colon of FMT recipients. (**E**) Microscopic examinations in H&E staining revealed inflammation or neoplasms in the rectum and colon of FMT recipients. * *p* < 0.05; ** *p* < 0.01; *** *p* < 0.001.

**Figure 6 ijms-23-09689-f006:**
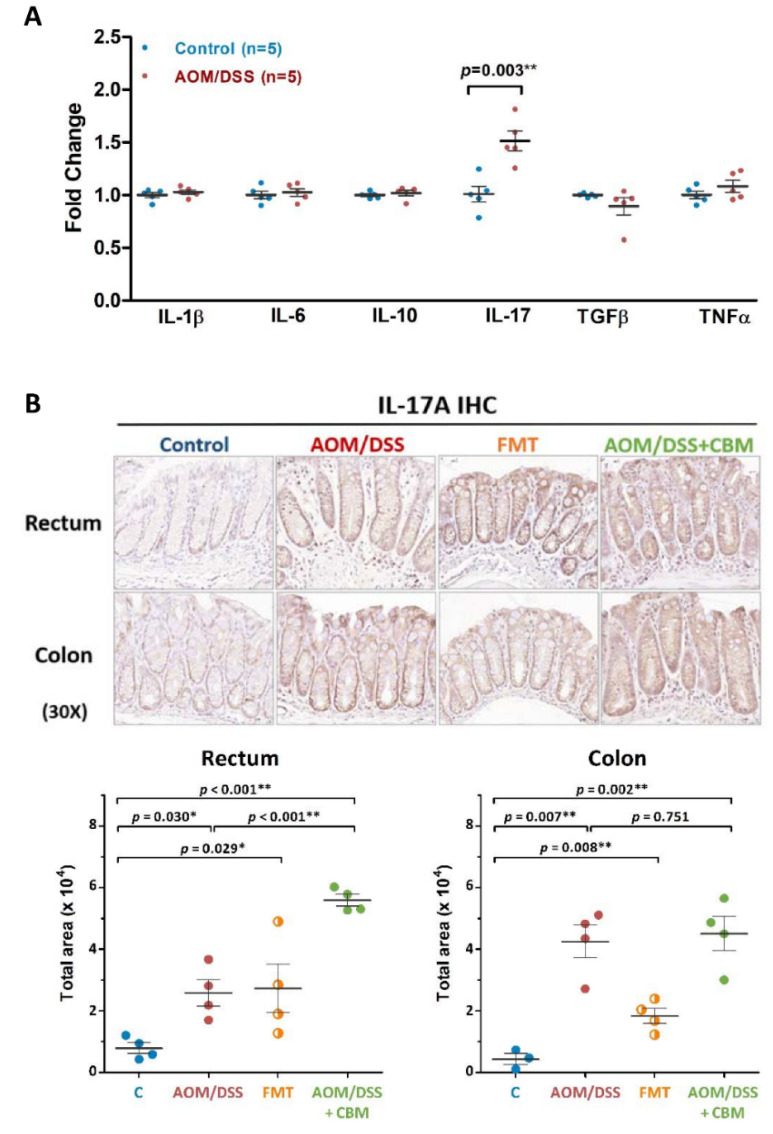
The expression of IL-17A in colon tissues from control, AOM/DSS, FMT, and AOM/DSS + CBM groups. (**A**) Screening the gene expression of in pro-inflammatory (IL-1β, IL-6, IL-17, and TNFα) and anti-inflammatory cytokines (IL-10 and TGFβ) in colon tissues using qPCR. Up-regulated IL-17 expression (*p* = 0.003) was observed in the AOM/DSS group when compared to the control group. The expressions of IL-1β, IL-6, IL-10, TGFβ, and TNFα were similar between the two groups. (**B**) Further validation using IL-17A IHC staining demonstrated a predominant expression of IL-17A in AOM/DSS, FMT, and AOM/DSS + CBM groups but not in the control group. The IL-17A-positive areas were calculated using ImageJ software. In the AOM/DSS + CBM group, the expression of IL-17A was significantly higher than in AOM/DSS group in the rectum (*p* < 0.001). * *p* < 0.05; ** *p* < 0.01.

**Table 1 ijms-23-09689-t001:** Results of probiotic interventions on murine colitis model in the literature.

Probiotic	Intervention	Murine Model	Result	Reference
*Clostridium* *butyricum*	Oral, 78 days 2 × 10^8^ CFU, 3 times a week	AOM/DSS	Decreased incidence and size of tumor Decreased TNF-α, IL-6, COX-2	[31]
*Clostridium* *butyricum*	Oral, 40 days 1 × 10^8^ CFU, 3 times a week	AOM/DSS	Lower tumor volume Lower IL-6, higher IL-10	[32]
*Lactobacillus* *bulgaricus*	Oral, 3 times for one week Pre-administration 1 × 10^9^ CFU/time	AOM/DSS	Inhibited tumor volume Decreased IL-6, TNF-α,IL-17, IL-23, and IL-1β	[33]
*VSL#3*	Oral, 12 weeks 1.5 × 10^9^ CFU/day	AOM/DSS	Decreased tumor load Decreased IL-6, TNF-α	[34]
*Escherichia coli Nissle 1917*	Oral, different courses: 7–14 days with or without pre-administration Different doses: 10^7^ and 10^9^ CFU/day	TNBS	Pre-administration and low dose (10^7^ CFU) protected colitis. However, pre-administration and high dose (10^9^ CFU) deteriorated colitis	[35]
*Lactococcus* *lactis*	Oral, 3 days Pre-administration Different doses: 1, 5, 10, 15 and 20 mg once daily	DSS	Probiotic deteriorated colitis, Increased IFN-γ, TNF-α, IL-6. Higher dose probiotic tended to decrease survival	[36]

Abbreviations: CFU, colony forming unit; AOM, azoxymethane; DSS, dextran sodium sulfate; TNBS, trinitrobenzene sulfonic acid.

## Data Availability

The datasets used and/or analyzed during the current study are available from the corresponding author on reasonable request.

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
