# Peer review of "Presume Why Probiotics May Not Provide Protection in Inflammatory Bowel Disease through an Azoxymethane and Dextran Sodium Sulfate Murine Model"

_ijms, 2022, doi:10.3390/ijms23179689_

Round 1

Reviewer 1 Report

This is a very well-designed and executed study with very interesting and provocative results. According to the results of the article, in an animal model of induced inflammatory bowel disease, a probiotic treatment containing a strain used with an intermittent sign of inflammation does not improve, but rather worsens the inflammation and has a carcinogenic role.

Regarding the Discussion, revision is suggested inserting the following aspects:

- it should be emphasized in the discussion that intermittent probiotic dosing was used

- a probiotic containing only 1 strain was used (CBM)

- No probiotic preconditioning was performed prior to AOM/DSS treatment.

- animals did not receive anti-inflammatory treatment in addition to the probiotic.

The article also raises additional questions, and these should be emphasized:

1., what if different strains/strains were used?

2. What effects would the use of anti-inflammatory treatment in addition to probiotic treatment have in the case of preconditioning probiotics?

After revision I suggest accepting the manuscipt for publication. 

Author Response

Response to Reviewer 1 Comments

Thanks for Reviewer 1 reading and providing valuable recommendations. Here, we appreciate and will reply and answer Reviewer 1’s recommendations and questions as follows. All the changes we made are marked in red text.

Point 1: Regarding the Discussion, revision is suggested inserting the following aspects:

- it should be emphasized in the discussion that intermittent probiotic dosing was used

- a probiotic containing only 1 strain was used (CBM)

- No probiotic preconditioning was performed prior to AOM/DSS treatment.

- animals did not receive anti-inflammatory treatment in addition to the probiotic.

Response 1:

Thanks for reminding to emphasize the study design in Discussion, let readers easily and well understand our study. Therefore, we added some sentences to explain our study design again in Discussion (please see Page 10, Line 277-279).

“In our study design, only one strain and non-preconditioning probiotic intervention was administered intermittently on AOM/DSS mice. No other anti-inflammatory medical treatment was given in additional to CBM.”

Point 2: The article also raises additional questions, and these should be emphasized:

1., what if different strains/strains were used?

  1. What effects would the use of anti-inflammatory treatment in addition to probiotic treatment have in the case of preconditioning probiotics?

Response 2: 

  1. Indeed, different strains may have different results. Thus, we reviewed related researches in the literature and mentioned these studies and results (please see Page 10-11, Line 275-298).

“Inconsistent results with our study, two studies showed Clostridium butyricum reduced colitis associated CRC on AOM/DSS murine models and decreased proinflammatory cytokines TNF-a, IL-6, and COX-2, and increased anti-inflammatory cytokine IL-10 [31,32]. Except CBM on AOM/DSS model, a study from Silveira et al. showed another strain intervention using Lactobacillus bulgaricus on AOM/DSS mice inhibited tumor volume and decreased pro-inflammatory cytokines IL-6, TNF-α, IL-17, IL-23 and IL-1β [33]. Another study from Wang et al. showed multi-strain probiotic intervention using VSL#3, which contained eight probiotic strains including Lactobacillus paracasei, Lactobacillus plantarum, Lactobacillus acidophilus, Lactobacillus delbrueckii subsp. bulgaricus, Bifidobacterium longum, Bifidobacterium breveCM, Bifidobacterium infantis, and Streptococcus thermophilus, altered the gut microbiomes and reduced the tumor load of AOM/DSS mice. In their study, they also put anti-inflammatory medicine, 5-ASA, for further analysis. They found both AOM/DSS mice treated with VSL#3 plus 5-ASA and VSL#3 alone improved colitis, but without head- to- head comparison between these two groups [34]. In addition, some studies showed different probiotic doses causing different effects. A study from Sha et al. showed pre-administered and low dose of probiotic intervention (107 CFU/day) using Escherichia coli Nissle 1917 on trinitrobenzene sulfonic acid (TNBS) treated mice significantly improved colitis. However, pre-administered and high dose (109 CFU/day) failed to improve colitis but deteriorated [35]. Another study from Komaki et al. showed Lactococcus lactis intervention on DSS mice deteriorated colitis, increased IFN-g, TNF-a and IL-6. Furthermore, mice on higher dose tended to decrease survival [36]. The results of the probiotic interventions on mouse colitis model were summarized in Table 1. Thus, the conflicting and uncertain results about the benefit of probiotic intervention are necessary to be clarified.”

Furthermore, we made a table to summarize these study designs and results (please see Table 1, Page 10, Line 254).

  1. Thanks for reviewer’s question. Due to inadequate evidence and researches about the issue, we are also curious about the effect about adding anti-inflammatory treatment to probiotics. In the study of Wang et al., “they also put anti-inflammatory medicine, 5-ASA, for further analysis. They found both AOM/DSS mice treated with VSL#3 plus 5-ASA and VSL#3 alone improved colitis, but without head- to- head comparison between these two groups.” We added the mention in Page 11, Line 286-289.

Thanks again for your review and valuable recommendations.

Reviewer 2 Report

The authors focused on presuming why probiotics may not provide protection in inflammatory bowel disease through an azoxymethane and dextran sodium sulfate murine model. Extensive and good Results part. Please see my suggestions.

1.     L93-96. Please make the AIM of the study the last, separate paragraph of the Introduction section, to be easier visible for the potential interested readers. Furthermore, please develop it more. Responding to the following questions would be helpful: What makes special this study? Which is its novelty character or its special aspects? What differentiate this paper from others in the same/similar topic?

2.     As I said, good Results part. However, the figures should be widened across the entire width of the page, in their current form being difficult to read them. The MDPI draft allows the presentation of large figures on the entire width of the page.

3.     Discussion section:

-       Please discuss previous studies where probiotics administration has had negative effects in vivo or in vitro. 

-       A summarizing table would be helpful. 

-       Further, discuss the physio pathological interaction between the inflamed colonic mucous and the substances produced by bacteria in the probiotic supplement. 

-       Have been your results observed with other bacteria species or there is data in literature where certain bacteria have shown benefit in IBD? I suggest checking and referring to https://doi.org/10.3390/diagnostics11061090

-       Detail if certain bacteria have superior beneficial role compared to other species. I suggest checking and referring to https://doi.org/10.3390/microorganisms9030618

-       I suggest making paragraph L350-352 the Conclusions section, and developing it a little more. Also, removing “In conclusion” from the beginning of the paragraph, as it is obvious and would be repetitive.

-       Instead of the actual paragraph L350-352, I suggest another, mentioning the strengths and the weakness of your study.

4.     I suggest tabulating paragraph  L314-421, it would be easier to follow it.

Author Response

Response to Reviewer 2 Comments

Thanks for Reviewer 2‘s comments and providing valuable suggestions. Here, we appreciate and reply to Reviewer 2’s suggestions as follows:

Because the original manuscript has been changed, the number of lines in the article also changed but without interfering the understanding. All the changes we made are marked in red text.

Point 1. L93-96. Please make the AIM of the study the last, separate paragraph of the Introduction section, to be easier visible for the potential interested readers. Furthermore, please develop it more. Responding to the following questions would be helpful: What makes special this study? Which is its novelty character or its special aspects? What differentiate this paper from others in the same/similar topic?

Response 1:

We added an AIM section (please see page 2, line 96-102) following the Introduction section and developed it more based on the reviewer’s valuable opinions as the follows:

2. Aim

We hypothesized probiotics or FMT in the condition of severe colitis with loss of mucosal barrier function such as moderate or severe IBD could not adhere to mucosa to provide protection in gut health. To simplify the reciprocal interactions between transplanted microbiomes and host gut and to avoid the complex and uncertain microbiomes of FMT, we tried to intervene single specific probiotic on AOM/DSS mice to explore the probiotic role in gut pathology, gut microbiota, and host immunity on a simulated IBD model.”

Point 2. As I said, good Results part. However, the figures should be widened across the entire width of the page, in their current form being difficult to read them. The MDPI draft allows the presentation of large figures on the entire width of the page.

Response 2:

Thanks for reminding the difficulty in reading these figures. We have widened these figures to read more easily.

Point 3. Discussion section:

-Please discuss previous studies where probiotics administration has had negative effects in vivo or in vitro.

-A summarizing table would be helpful. 

Response:

Some studies showed negative results of probiotic intervention on mouse colitis model. We have mentioned these studies in Discussion section and made a table to summarize these study designs and results (please see page 11, line 275-298). We have made a table to summarize the results of related studies (please see Table 1. page 10, line 254).

“Inconsistent results with our study, two studies showed Clostridium butyricum reduced colitis associated CRC on AOM/DSS murine models and decreased proinflammatory cytokines TNF-a, IL-6, and COX-2, and increased anti-inflammatory cytokine IL-10 [31,32]. Except CBM on AOM/DSS model, a study from Silveira et al. showed another strain intervention using Lactobacillus bulgaricus on AOM/DSS mice inhibited tumor volume and decreased pro-inflammatory cytokines IL-6, TNF-α, IL-17, IL-23 and IL-1β [33]. Another study from Wang et al. showed multi-strain probiotic intervention using VSL#3, which contained eight probiotic strains including Lactobacillus paracasei, Lactobacillus plantarum, Lactobacillus acidophilus, Lactobacillus delbrueckii subsp. bulgaricus, Bifidobacterium longum, Bifidobacterium breveCM, Bifidobacterium infantis, and Streptococcus thermophilus, altered the gut microbiomes and reduced the tumor load of AOM/DSS mice. In their study, they also put anti-inflammatory medicine, 5-ASA, for further analysis. They found both AOM/DSS mice treated with VSL#3 plus 5-ASA and VSL#3 alone improved colitis, but without head- to- head comparison between these two groups [34]. In addition, some studies showed different probiotic doses causing different effects. A study from Sha et al. showed pre-administered and low dose of probiotic intervention (107 CFU/day) using Escherichia coli Nissle 1917 on trinitrobenzene sulfonic acid (TNBS) treated mice significantly improved colitis. However, pre-administered and high dose (109 CFU/day) failed to improve colitis but deteriorated [35]. Another study from Komaki et al. showed Lactococcus lactis intervention on DSS mice deteriorated colitis, increased IFN-g, TNF-a and IL-6. Furthermore, mice on higher dose tended to decrease survival [36]. The results of the probiotic interventions on mouse colitis model were summarized in Table 1. Thus, the conflicting and uncertain results about the benefit of probiotic intervention are necessary to be clarified.”

- Further, discuss the physio pathological interaction between the inflamed colonic mucous and the substances produced by bacteria in the probiotic supplement.

-Have been your results observed with other bacteria species or there is data in literature where certain bacteria have shown benefit in IBD? I suggest checking and referring to https://doi.org/10.3390/diagnostics11061090

-Detail if certain bacteria have superior beneficial role compared to other species. I suggest checking and referring to https://doi.org/10.3390/microorganisms9030618

Response: Thanks for reminding the discussion about the aspect of physiopathological interaction between the inflamed colonic mucous and the substances produced by bacteria in the probiotic supplement. We learned a lot from the two recommended references and added the part of the physiopathological interaction between colitis and microorganisms to enrich the Discussion section (please see page 10, line 257-264).

“From the pathophysiological point of view, altered gut microbiota in IBD can be explained due to the increase of some bacteria which induce some proinflammatory cytokines and the reduction of other bacteria which induce anti-inflammatory cytokines and product beneficial metabolites, thereby disturbing the gut immune homeostasis, and decreasing the gut protection. Thus, the treatment effect should be achieved if the intervention of good live microorganisms can be adhering to the intestinal epithelium, producing beneficial metabolites, stabilizing the intestinal microbiota, and stimulating anti-inflammatory cytokines in the gut environment [29,30].”

Furthermore, Akkermansia muciniphila (A. muciniphila) is considered a promising probiotic strain in the future and plays a vital role not only in gut health but other physiopathological conditions such as obesity, diabetes mellitus and metabolic syndrome. We also added the introduction and discussion of A. muciniphila in the Discussion (see page 11, line 299-306).

“Nowadays, accumulating evidences have shown Akkermansia muciniphila (A. muciniphila) plays an important role in gut health. A. muciniphila, colonizing the human gut and accounting for 3-5 % of gut microbiomes, was first isolated in 2004 by Derrien et al. and has been found with various benefits in obesity, diabetes mellitus and metabolic syndrome in addition to gut health [37,38]. Some studies revealed lower colonization and abundance of A. muciniphila in IBD patients and mouse colitis models. Intriguingly, either live or pasteurized A. muciniphila improved colitis in mice [39,40]. A muciniphila has been considered a promising probiotic strain in the future and is worth for further study.”

- I suggest making paragraph L350-352 the Conclusions section, and developing it a little more. Also, removing “In conclusion” from the beginning of the paragraph, as it is obvious and would be repetitive.

- Instead of the actual paragraph L350-352, I suggest another, mentioning the strengths and the weakness of your study.

Response: Thanks for your suggestion. We made the Conclusion section and developed it more by mentioning our opposite results, and explaining the limits of our study (please see page 12, line 344-350).

5. Conclusion

Probiotic intervention in IBD patients may not always provide protection. To some extent, they can cause dysbiosis and elicit further inflammation. There are still some limits in our study and it is worth investigating the treatment effects in different strains, durations, doses, frequencies of probiotic intervention with or without precondition. The role of probiotics in IBD should be explored more and probiotics in IBD patients should be used more cautiously.”

Point 4. I suggest tabulating paragraph L314-421, it would be easier to follow it.

Response 4: Thanks for suggestion. In addition to mentioning these results of related studies in the text, we also made a table to summarize the design and result in each study (see Table 1., page 10, line 254). 

Thanks again for your review and valuable recommendations.

Round 2

Reviewer 2 Report

The authors responded properly almost to all my requests, excepting:

I requested in my previous report: Please make the AIM of the study the last, separate paragraph of the Introduction section (not a separate section!!!), to be easier visible for the potential interested readers. So, please put the aim of the study in its proper place.

Please check again the order of the sections , requested for an original article, by the instructions for authors at the link https://www.mdpi.com/journal/ijms/instructions Conclusions section must be AFTER the Materials and Methods section.

Author Response

Response to Reviewer 2 Comments

We thank and appreciate for Reviewer’s corrections.  

Comments and Suggestions for Authors

Point 1. The authors responded properly almost to all my requests, excepting:

I requested in my previous report: Please make the AIM of the study the last, separate paragraph of the Introduction section (not a separate section!!!), to be easier visible for the potential interested readers. So, please put the aim of the study in its proper place.

Response: Thanks for reviewer’s correction. We have made the aim of the study in the last and separated paragraph of the Introduction section.

Point 2: Please check again the order of the sections, requested for an original article, by the instructions for authors at the link https://www.mdpi.com/journal/ijms/instructions Conclusions section must be AFTER the Materials and Methods section.

Response: Thanks for reviewer’s reminder. We apologized for making such an error and corrected the order of the Conclusions section after Materials and Methods section.

Thanks again for reviewer’s corrections and recommendations.
